# Relationship of Genetic Polymorphisms and Microbial Composition with Binge Eating Disorder: A Systematic Review

**DOI:** 10.3390/healthcare12141441

**Published:** 2024-07-19

**Authors:** Montserrat Monserrat Hernández, Diana Jiménez-Rodríguez

**Affiliations:** 1Department of Geography, History and Humanities, University of Almería, 04120 Almería, Spain; 2Department of Nursing, Physiotherapy and Medicine, University of Almería, 04120 Almería, Spain; djr239@ual.es

**Keywords:** compulsive eating, genes, polymorphism, binge eating disorder, microbiota, psychosocial factors, precision nutrition

## Abstract

Humans are the result of an evolutionary process, and because of this, many biological processes are interconnected with each other. The intestine–brain axis consists of an intricately connected neuronal–neuroendocrine circuit that regulates the sensation of hunger and satiety. Genetic variations and the consumption of unnatural diets (ultra-processed foods, high contents of sugars, etc.) can override this circuit and cause addiction to certain foods and/or the inability to feel satiety in certain situations. The patients who come to consultations (mainly psychology or nutrition) in an attempt to resolve this problem sometimes fail, which leads to them looking for new strategies based on biological predisposition. This investigation aims to evaluate the genetic studies regarding the microbiota carried out in the last 12 years in humans to try to determine which genes and microbes that have been recently studied are related to patients diagnosed with binge eating disorder or compulsive eating (presenting obesity or not). The protocol followed the PRISMA statement, and the following databases were searched from 2012 until the present day: PubMed, PsycINFO, SCOPUS, and Web of Science. Twenty-four international articles were analyzed, including cross-sectional or exploratory studies; five of them referred to the microbial composition, and in nineteen, the existence of genetic polymorphisms present in binge eating disorder or in compulsive eating could be observed: DRD2, OPRM1, COMT, MC4R, BNDF, FTO, SLC6A3, GHRL, CARTPT, MCHR2, and LRP11. Even though there is still much to investigate on the subject, it must be highlighted that, in the last 4 years, a two-fold increase has been observed in potential markers and in studies related to the matter, also highlighting the importance of different analyses in relation to psychosocial factors and their interaction with the genetic and microbial factors, for which research on the matter must be continued.

## 1. Introduction

The main eating disorders (EDs) described by the Diagnostic and Statistical Manual of Mental Disorders (DSM-V) are anorexia nervosa (AN), bulimia nervosa (BN), and binge eating disorder (BED) [1]. AN is characterized by a restriction in the consumption of energy with respect to the necessary requirements, a significant low weight according to the age, sex, stage of development, and physical health of the patient, fear of gaining weight, and distortion of reality with respect to weight; BN consists of recurring episodes of binge eating (at least once a week for 3 months), followed by compensatory behaviors (vomiting, laxative and/or diuretic abuse, intense exercise, or fasting); and for BED, we find recurring (≥1 times per week, for 3 months minimum), brief (≤2 h), and distressing binge eating periods, during which the patients feel a loss of control, over which they consume large quantities of food as compared with the majority of people in similar circumstances [1].

In general, they share an alteration in feeding behavior [2], for which it can be normal for individuals affected to acquire a spectral functionality, fluctuating between those who suffer from it (a person with AN can also have BN and/or BED) [1]. Additionally, disorders instigated by substances (alcohol and tobacco consumption, etc.) are more prevalent in patients with eating disorders than in the general public, so it is assumed that it is a common intrinsic component linked to addiction (presently, the genetic and neurological factors are the most investigated) [3]. The causes that lead to the development of EDs are linked to biological factors, social factors, and personal development [2]. Regarding biological factors, genetics play a fundamental role in the manifestation of these pathologies. In fact, genetics predispose between approximately 33% and 84% of AN, between 28% and 83% of BN, and between 41% and 57% of BED [4].

Despite it not being as well researched, BED is the most prevalent within Western society [5], with a prevalence ranging from 2% to 4% of the population aged 16–20 years compared with AN (0.8–2%) and BN (2–3%) [4]. In addition, it may be under-diagnosed. Patients have an elevated risk of being overweight or obese, along with the related consequences, hence resulting in difficulties in their physical and psychological health, with those suffering from it sometimes not having or being unable to access the necessary tools to find solutions [6,7]. In May 2013, the American Psychiatry Association (APA) officially recognized BED as a different eating disorder, with a lower diagnostic threshold (in frequency and duration of symptoms) than those previously established, for which it is expected that the number of diagnosed patients with a need for treatment will increase [1,8,9].

The treatment aims to decrease the frequency of binge eating periods, as well as cognitive impairments associated with malnutrition, improve metabolic health and weight loss (in patients who are overweight or obese), and regulate their well-being. The main focus of the treatment includes psychological interventions (cognitive behavioral therapy, dialectical behavior therapy, interpersonal psychotherapy, and behavioral weight loss therapy), pharmacological treatments (antidepressants, anticonvulsants, anti-obesity agents, and central nervous system stimulants), or a combination of both [10]. Nevertheless, they are not sufficient for decreasing its incidence or prevalence. Some of the literature that currently exists describes the existence of candidate genes linked to EDs through genome-wide association studies (GWASs) that seek to find the causes in order to improve the treatment. From Klein’s and collaborators’ studies [11], ~4000 GWASs in humans have examined ~2000 diseases and traits [12]. Also, epigenetic studies of gene expression and interactions between genes and nutritional genomics are being promoted [13]. However, more research on AN and BN currently exists as compared with research on BED.

Nevertheless, despite the different open branches in the field of genetics, these studies do not offer complete solutions for the existence of EDs. Other investigations stated that there is a correlation between the microbiome, appetite, and body weight [14,15,16]. Therefore, and based on the development of the Human Microbiome Project [17], genes and microbiome genomes located in the mouth, intestine, vagina, and skin that may have a relation to this type of disorder have been found and described.

Thus, based on the two major studies mentioned above, the Binge Eating Genetics Initiative (BEGIN) [18] emerged as a multidimensional study examining the interaction of the genome with the gut microbiota, as well as phenotypic data, to provide treatment responses for BED and BN.

Despite the advancements in the field of biomedicine, the causes of EDs in general and BED in particular have not been completely explained. This is because EDs (and BED) have a high social interaction (familiar and similar groups of people), which affects both the present and how the person develops until they become an adult [19], which influences the acceptance (or not) of the sensorial characteristics of food/dishes; economic and ecological factors; their perception of food and how it is classified; symbolic factors associated with them with respect to elements of social status, gender, age, beliefs, knowledge, and assigned values; as well as the relationship between health, image, and/or aesthetics [20,21].

Through the present review, we aim to propose a multifactorial biological disease model based on current genetic findings and the state of the microbiome, in addition to psychosocial aspects. Due to all of this, through the present article, we hope to summarize the results of the studies on the genetic determinants of BED (including the serotonergic genes, dopaminergic genes, opioids, appetite regulators, and endocannabinoids), as well as factors related to the microbiome and the psychosocial factors, in the last few years, with the aim of developing the hypothetical BED. We do not claim to provide detailed information about genetic, analytic, and neurological methods, as in the past few years, they have been further developed as research advances [5,22,23,24,25,26,27]. We do find interesting the elaboration of an analysis pattern bearing in mind the maximum possible factors that influence patients with BED.

## 2. Materials and Methods

### 2.1. Protocol and Registration

The present study was conducted by following the recommendations of the Preferred Reporting Items for Systematic reviews and Meta-Analyses statement (PRISMA) [28] for genetic and microbiota factors. It is registered on the International Prospective Register of Systematic Reviews website (PROSPERO-ID: CRD42024545414) (https://www.crd.york.ac.uk/prospero/display_record.php?RecordID=545414, accessed on 30 May 2024).

### 2.2. Information Sources and Search Criteria

To identify potentially relevant investigations, searches in various databases were performed for articles published in the last 12 years (February 2012–2024); including PubMed, Scopus, PsycINFO, and Web of Science. The following keywords were used, which referred to biological factors: (binge eating disorder OR compulsive eating) AND (genes OR genotypes OR polymorphism); (binge eating disorder OR compulsive eating) AND (microbiome OR microbiota). Due to the high psychological and social component, a search for information with the following terms was also conducted: (binge eating disorder OR compulsive eating) AND (genes OR genotypes OR polymorphism) AND (psychological factors OR psychosocial development OR psychosocial factors) to obtain sufficient information to contribute to the classification and treatment of this disease. After obtaining all the potential publications, filters were applied with respect to studies in humans and scientific articles, after which a process of selection (2 reviewers) took place to clearly determine if the criteria of inclusion were met: (1) examine title and abstract and (2) completely reading the study to ensure that compatibility existed between the criteria of research. If both reviewers could not come to a consensus, the information would be provided to an expert (an external reviewer) to contribute to the final decision. Articles that were not related to BED and social factors were excluded, as well as treatment studies (due to the insufficiency of the studies with respect to the microbiota, the information provided in articles in this respect was accepted, in which cross-cutting data were offered, as well as treatment results). Moreover, the articles found in the bibliography were of interest to this study (as in the case of specific polymorphisms or a review of previous years when information of relevance was observed). The final results were recorded in a document (EXCEL sheet) divided into three sections (genetic factors, microbiota, and psychosocial factors); the following information concerning every article was available to each member of the team: author, publication year, characteristics of the sample, polymorphism of the candidate genes and/or microbiota aspects analyzed and/or social factor analyzed, methodology, primary results, and conclusions.

### 2.3. Eligibility Criteria

Eligibility criteria (all of them must be met): (a) directly measured the relationship between variables (genetics, microbiota, or psychosocial variables) and BED; (b) having empirical quantitative data (qualitative analyses were excluded, as well as revisions and commentaries); (c) published in English.

### 2.4. Risk of Biases

To avoid risks of biases in the analysis, we evaluated each study according to the Strengthening the Reporting of Observational Studies in Epidemiology (STROBE) checklist [29] (0–21 points). This is a tool used to evaluate the quality of cross-cutting studies on health; it is composed of a verification checklist of 21 elements that facilitate the evaluation criteria in the interpretation of the studies.

## 3. Results

### 3.1. Genes, Polymorphisms, and BED

Figure 1 shows the selection process that was utilized. The initial search provided 697 records for genetic factors and BED, of which 19 were optimal for our analysis.

#### 3.1.1. Main Characteristics of the Analyzed Studies

The ages of the case and control subjects ranged from 14 to 65 years old. BED was primarily assessed based on the DSM-V criteria. In studies linked to compulsive eating, in which the diagnosis of BED was not used, binge eating control was taken into consideration through a medical report. The design of the studies was mainly case–control. Studies on minors were found in a low percentage, and in the case of any, they were part of a larger-scale longitudinal research study that involved entire families. Studies on women were predominant, and in those that included men, women were found in percentages greater than 80%. Within the selected studies, a total of 14 genes were directly linked with BED, and 3 associated with binge eating episodes (HTR2A, MCR4, and OXTR) were analyzed. The characteristics of the 17 articles included in this review are summarized in Table 1.

#### 3.1.2. Study Results

##### Dopaminergic Genes

Subjects with BED are significantly associated with the rs1800497 and rs6277 SNPs of the DRD2 and DRD3 Ser9Gly genes [30,42,46]. And current studies show an increased prevalence of BED in the significant presence of the BDNF (rs6265) and DRD2 (rs1800497) gene SNP combination.

Specifically, recurrent binge eating episodes in BED patients are associated with low levels of dopaminergic transmissions, especially linkages with DRD2, Taq1A, DrD47R, and COMT genes [34,35,37,45].

However, there is research showing non-significant relationships between dopaminergic genes and BED, when the cause is sought in the complex etiopathology [46].

##### Serotoninergic Genes

Munn-Chernoff et al. [3] found no significant associations between any genetic variants of serotoninergic and BED genes. Koren et al. [32] showed that the presence of the SNPs rs6561333 and rs2296972 could have a protective influence against binge eating (however, this study did not test patients diagnosed with BED).

##### Other Genes

Although not yet confirmed for BED, significant associations have been observed between the G-T-A-G haplotype of the OXT gene and the preference for eating foods rich in fats and sugars and, therefore, having a particular sensitivity to binge eating disorders [36].

Regarding the FTO gene and its influence on binge eating, we can observe, on one hand, studies that show significant relationships (SNP rs1421085 and rs1121980) [38] and, on the other hand, those that do not [40]. However, there is a confirmed relationship between this gene, leptin, and increased appetite [40].

Studies on the MCR4 gene (rs17782313) have observed associations with severe binge eating without a diagnosis of BED [44].

Although there are not yet many association studies between FAAH and binge eating, a higher frequency of the A allele is observed in women with BED [41].

### 3.2. Microbiota and BED

Figure 2 shows the process of selection. The initial research provided 168 records for microbiota and BED, from which only 5 were optimal for our analysis.

#### 3.2.1. Main Characteristics of the Analyzed Studies

Every analyzed control–case assay and study were carried out in subjects older than 18 years old, with a high percentage of people diagnosed with obesity or high BMI, including those with bariatric surgery. The BED was assessed based on the DSM-IV andDSM-V criteria. In the studies related to eating compulsively, where a BED diagnosis was not used, binge eating control was taken into account through medical forms. Studies in women were predominant. The design of the studies was diverse; these were: case–control (2), cohort (1), and experimental (2) (see Table 2). Also, due to the small amount of literature regarding this topic, for the discussion and comparison, five systematic reviews on the associated literature were taken into consideration.

The results cannot be compared among themselves due to the difference between the type of intervention (in intervention studies) and/or analysis performed (see Table 3).

#### 3.2.2. Results from the Studies

In addition to the studies shown in Table 2, in the following section, we refer to other reviews from previous years, where we describe the relevant data found.

##### Bacteria

Decreases in *Akkermansia* and *Itestimona* and increases in *Bifidobacterium*, *Roseburia*, and *Anaerostipes* have been described in obese patients with BED [50]. These data are yet to be further explored; however, there is corroborated evidence of the relationship between intestinal dysbiosis and BED [49]. Dysbiosis also exists in AN, but with a different proportion of bacteria than that observed in BED [52]. The data that can be extracted, and which are of interest to our research, are that binge/purge and restriction are related to different compositions of the gut microbiome.

To date, there has been only one cross-sectional study specifically relating BED to microbiota composition [50]. It proposes that microbiota homeostasis is essential for a healthy gut–brain, so microbial imbalance is observed in all EDs [53].

##### Microbiota–Intestine–Brain Relationship

Gut dysbiosis is associated with a reduction in the tryptophan-related neuroprotective metabolite, indole propionate, and increased communication between certain reward regions of the brain [49]. In addition, elevated concentrations of ClpB (produced by *Esche-richia coli*) are observed in patients with ED and positive correlations with scores on the EDI-2 questionnaire [47].

In the review by Herman and Bajaka [54], it was revealed that stress was the promoter of binge eating in patients with BED, so it seems that, in these patients, the reward system is more strongly activated by food than in obese people without BED. Serotonin, dopamine, noradrenaline, and acetylcholine produced by gut bacteria may influence this process.

In the review by Hayatte-Dounia [55], the important role of the endocannabinoid–gut microbiome axis in ED is highlighted, without being specific to BED, but proposing that endocannabinoids play a key role in the regulation of food intake.

##### Consumption of Probiotics and Medication

It has been observed that patients diagnosed with BED have received antimicrobial treatment more frequently compared with control patients [48].

The consumption of Bifidobacterium lactis (Bi-07) in one tablet for 90 days manifests in a reduction in food addiction symptoms and binge eating scores [51]. And supplementation with synbiotics (prebiotics + probiotics) is associated with improvements in microbial diversity, attenuating inflammatory responses, possibly associated with this disorder [56].

Figure 3 shows an overview of the compiled information based on genetic factors and microbiota and its influence on BED.

## 4. Discussion

The aim of the present work was to provide information, as clearly and reliably as possible, on genetic polymorphisms and the state of the microbiota related to BED through an analysis of the scientific literature, for it to be useful in the search for professional interventions on BED. The main results show that only a small number of polymorphisms have been analyzed for each gene until now, with a direct relationship with BED (ABCA1 rs9282541; BDNF rs6265; DRD2 rs1800497; DRD3 Ser9Gly; DRD4 rs936461; FTO rs9939609; and Taq1A rs1800497), although there has been a larger number of articles related to episodes of compulsive eating manifested in other EDs, such as AN or BN (COMT Val 108/158; HTR2A rs1923882, rs6561333, and rs2296972; FTO rs1421085, rs1121980, and rs1558902; MCR4 rs17782313; and OXTR rs2254298 and rs53576). Also, in BED, a situation of intestinal dysbiosis was found, with decreases in Akkermansia and Itestimonas and increases in Bifidobacterium, Roseburia, and Anaerostipes, although it was not clarified if this was due to BED or the obesity manifested. The ingestion of Bifidobacterium lactis for 90 days decreased the symptoms of BED and the addiction to food in the patients analyzed [51].

### 4.1. Genes and BED

In general, BED is manifested with obesity, and the tendency to eat compulsively can be influenced by hyper-reactivity to the medicinal properties of foods (predisposition that is easily exploited in our surroundings) with highly visible effects and easy access to sweet foods, fatty foods, and ultra-processed foods [57]. No GWASs were found that specifically analyzed BED, as compared with AN or BN [57,58,59,60,61,62,63,64]; only analysis studies of partial genes were found.

The most studied gene was FTO [24,33,38,39,40,65,66], associated with fat mass. The protein fat-mass-and-obesity-associated (FTO) is present in various metabolic active tissues: the heart, kidneys, fatty tissue, and brain. In particular, it is strongly expressed in the hypothalamus, related to the regulation of the system of excitement and appetite (as well as temperature regulation, autonomous function, and endocrine function). This is why it plays an essential role in the feeling of satiety, as evidence exists of the over-expression of the FTO gene leading to an increase in the intake of foods (especially with high percentages of sugars and fat) [33], as well as feeling of anxiety toward food [38]. It is also related to BMI [33,40]. Moreover, a connection with the threshold of detection of proteins has been found, for which, with the same concentration of amino acids in the intracellular medium, the stimulation of FTO in the carrier cells of allele A of SNP rs9939609 is less than that in the carrier cells of the genotype TT [39]. Nevertheless, conclusive results with respect to the aforementioned influence in direct connection to BED [39] do not exist, although it is found in patients with AN and BN with episodes of binge eating [67].

Even though there are no studies that directly relate BED to the proteins Taste Receptor Type 1 Member 2 (TAST1R2) and Taste Receptor Type 1 Member 3 (TAST1R3), which are expressed in the tongue and palate, their importance has been observed, as they are described to participate in the regulation of taste (specifically in the taste of sweet flavors), and recent studies about obesity have proposed that the increase in the consumption of sweet foods can be affected by allele G of rs120338082 of the TAST1R2 gene, which codes for a protein that increases the threshold of sweet flavors [67,68].

The Glucose Transporter 2 (GLUT2) protein allows glucose to exit the enterocytes of the intestine to reach the bloodstream, to later enter liver and kidney cells. Due to its low affiliation for glucose and its great velocity, it has been proposed as a glucose sensor, and is considered very important in the regulation of food intake [65]. Individuals with the genotype TT for SNP rs5400 of the gene SLC2A2, which codes for GLUT2, are more likely to have higher sugar intake and an increased likelihood of predisposition to ultra-processed foods, derived from the under-detection of glucose in the blood [65,69,70]. Nevertheless, studies analyzing the relationship between this gene and BED do not exist.

Human genetics and studies in animals have suggested that changes in neurotransmitter networks exist, including the dopaminergic system, the serotoninergic system, and the opioids associated with binge eating [71]. The protein Midbrain Dopamine Circuits plays an important role in addiction and in behavior related to food, as it is involved in the signaling of the dopaminergic system through the Dopamine Receptor D2 (DRD2) receptor [72]. Specifically, the T allele of SNP rs180497 of the gene that codes for DRD2 is associated with a reduction in the density of DRD2 and a smaller number of dopamine attachment places in the cerebral striatum, in comparison to the genotype CC. This reduction makes the carriers less sensitive to the activation of the reward circuits that are based on dopamine, resulting in excess eating [30,45,46]. This allele is also associated with substance abuse disorders, hence the existence of the relationship previously described in the present work [72,73].

One of the functional polymorphisms of the receptor D2 studied with more frequency was Taq1A (Rs1800487), which was thought to be located in the untranslated region 3 of DRD2. However, previous studies have shown that this SNP was not a part of the DRD2, but part of the ANKK1 gene [74] in axon 8 and, additionally, that there was a connection with the dopaminergic system. Genetic variations in ANKK1 in patients with BED have been described [30,75], especially in allele G, which is associated with a significant risk of developing BED [30,37].

DRD3 is also a dopamine receptor gene. In current studies regarding EDs, an existing strong interaction effect between genotype DRD3 Ser9Gly and the diagnostic scales of BN [76] has been described. Specific combinations of SNP from DRD2 and DRD4 show a greater relationship with BED patients [77].

In the case of dopamine carrier genes, the relationship between BED and DAT (polymorphism 10/9R) is also being studied, as well as the fluctuation between AN and BED [42]. However, some studies have not found a relationship between the associated polymorphisms with dopamine and the observed behaviors in BED [43]. This is why there are still no studies that corroborate the proposed results, but rather argue for the search for more dopaminergic polymorphisms or other candidate genes that affect other neurotransmission systems that could have a greater influence on BED [43].

The SNP rs1799971 (polymorphism A118G) of the gene OPRM1 (opioid receptor mu 1) has been found with greater frequency in people diagnosed with BED [30,78]. Even though there are not sufficient bibliographic results in humans related to this, murine studies have shown that the group with BED had an allele G with a frequency of 18.5%, in comparison with 9.6% in obese mice without BED [79]. Human-based studies that investigated the opioid receptors in the brain described a general anomaly in the cerebral opioid function in these conditions, and therefore, the causes must be further investigated [78,79].

The hormones ghrelin and leptin play key roles in the regulation of appetite, the ingestion of food, and energy metabolism [80]. The protein Fatty Acid Binding protein 2 (FABP2) is found in the epithelial cells of the small intestine, where it has a major influence on fat absorption and its metabolism. Physiologically, it helps to decrease the feeling of hunger. Allele A of SNP rs799883 of the gene that codes for FABP2 is associated with higher levels of leptin, as well as resistance to insulin [23,27], and produces a greater hormonal activation towards eating compulsively.

The protein Circadian Locomotor Output Cycles Kaput 1 (CLOCK) is also considered interesting due to the resulting circadian variations, specifically by allele C of SNP rs1801260 of the gene that codes it, causing modifications in the levels of insulin, leptin, and ghrelin, which influence appetite and the times of ingestion [81].

Alterations in the neurotransmission of serotonin in patients with AN or BN with tendencies toward binge eating episodes have been identified [32]. For example, it has been found that the absence of tryptophan in the diet, a precursor of serotonin (5-HT), corresponds to greater ingestion of foods in people with BN [82]. They manifest normal levels of 5-hydroxyindoleacetic acid in the cerebrospinal fluid when symptoms improve [83]. This is why gene HTR2A has started to become a gene-of-interest within the scientific community when classifying the types of AN, and associations between promotor-1438G/A in the SNP Rs6311 in binge–purge AN and BN that manifests with binge eating [83,84] have been observed. Studies on this gene in BN and BED are limited, even though relationships between the SNPs rs6501333 and rs229697, and protection from binge eating, have been observed [32].

The gene that codes for the protein Melanocortin 4 receptor (MCR4R) belongs to a family of membrane receptors that activate the response against Melanocortin. This protein participates in the regulation of satiety after the ingestion of food at the level of the hypothalamic–pituitary axis through the interaction with leptin. Allele C of the SNP rs17782313 of the gene that codes for MC4R decreases the levels of the aforementioned protein, which puts the regulation of appetite at risk (increasing snacking and prevalence) [44]. A greater prevalence of binge eating in women with this allele has also been observed [44].

The BDNF protein is involved in the suspension of appetite, and the polymorphisms rs1691237 and rs6265 have been analyzed in the context of EDs, mainly AN and BED [85]. In the case of BED, the presence of rs6265 (GA + AA) is more frequent in patients with obesity, especially in the context of weight gain [46]. However, other studies do not show any associations [39].

The genes OXTR and FAAH seem to be related to binge eating processes; however, there are not enough studies that directly relate them to BED [36,41].

There are contradictory studies with respect to other genes, such as 5-HTTLPR, SLC6A4, or GHRL, which still do not show relevant results [31,39].

### 4.2. Microbiota and BED

Even though very few cross-cutting and observational studies exist regarding the relationship between the status of the microbiota and BED [25], in the present investigation, different studies were analyzed where common factors between the observed pathologies and our pathology of interest (BED) exist, for example, the existence of compulsive eating episodes [9,86].

In general, a clear dysbiosis in patients with ED was observed [25,52,58,87,88], specifically in BED [48]. Also, when modifications to the microbiota composition were performed, for example, through fecal transplantation (FMT) or the consumption of antimicrobial drugs for long periods of time, modifications to feeding behavior were observed [48,57,89]. It has been observed that microbiota related to obesity is rich in Lactobacillus reuteri and poor in Bifidobacterium animalis and Methanobrevibacter [58,60]. These data are consistent with a study on BED in humans, in which Akkermansia was found to be low and Bifidobacterium to be high [50]. Additionally, it seems that having a higher (obesity or overweightness) or excessively low BMI (as it occurs in AN) is predictive of dysbiosis [50,52,88,90]. This information does not provide results about the way of eating in BED, but it must be highlighted that a large majority of patients with BED are overweight and/or obese, and that the microbiota composition is related to appetite [24,25].

Nevertheless, the majority of studies in this respect have been carried out in murine models, and in addition to the extrapolation of the results being too precipitated, it must not be forgotten that the composition of the intestinal ecosystem is different from patient to patient; therefore, working individually on every bacterial strain can be an error, which could be reduced by analyzing the functionality of the ecosystem as a whole. In the study conducted by Visconti et al. (2019) [91], it was observed that subjects who were not related shared, on average, nearly double the metabolic pathways (82%) than the species (43%). It was also found that the metabolic pathways were associated with 34% of the blood metabolism and 95% of the fecal metabolism, with more than 18,000 significant associations, whereas the species showed less than 3000 associations.

Attention must be given to the relationship between the microbiota-intestine–brain–axis, because in patients with BED, stress plays an important role in binge eating [54]. Additionally, studies are being conducted on how indo-propionate metabolites related to tryptophan are reduced in processes of intestinal dysbiosis in BED, resulting in a reduction in the feeling of wellness [49]. Furthermore, this dysbiosis leads to alterations in the plasma concentrations of regulatory proteins. ClpBs are especially elevated in patients with EDs and can be associated with psychopathological features, such as anxiety [47]. It has also been observed that obese subjects, both with and without BED, are associated with a generalized reduction in the availability of the cerebral mu opioid receptor (MOR), which could be associated with the BMI factor [78]. In another study [92], it was observed that a reduction in fat consumption in men reduced the availability in the opioid receptor and an alteration in dopamine signalization. Another recent investigation showed that changes in the pro-inflammatory cytokines, levels of serotonin, and microbiota status, change someone’s mood (anxiety and depression) in eating disorders [93], with modifications observed when re-nutrition occurs in patients with ED [59]. Moreover, some studies exist that describe the relationship between serotonin and the intestine, especially in murine research, with the restructuring of the microbiota leading to improvements in mood [94,95].

With respect to the metabolic products of the intestinal bacteria, it has been found that the abundant existence of butyrate is inversely proportional to anxiety [47,92,96] and propionate has an inversely proportional relationship with the secretion of insulin [59,90,97], with both of these compounds related in binge eating and BED studies [96,98]. The amplification of colonic propionate in humans reduces the anticipatory reward to images of high-calorie foods in the caudate and NAcc brain regions in non-obese men [99], and supplementation with propionate prevents the increase in weight and the resistance to insulin in adults who are overweight [100], potentially due to the positive regulation of the signaling of the GLP-1 receptor. Short-chain fatty acids (SCFAs) can also go through the blood–brain barrier to act directly on the hypothalamic neurons and act on the suppression or activation of appetite in the neuronal pathways in the hypothalamus. In addition, in studies about ED, an increase in glutamate in the neurons POMC/CART (suppressor pathway of appetite) and a decrease in the transmission of GABA in the neurons NPY/AGRP (activation pathway of appetite) were observed [101].

Furthermore, intestine permeability as a consequence of a poor microbial composition increases the permeability of the blood–brain barrier through an increase in the plasmatic levels of free fatty acids and an increase in the production of ketone bodies with states of depression and/or anxiety [102].

Even though only a few studies exist regarding interventions with probiotics, most of which are based on AN [56], some of them seem to show that supplementation with Lactobacillus and Bifidobacterium can decrease binge eating [51], while other Bifidobacteria can be considered as activators of BED [50].

### 4.3. Relationship between Environmental Factors, Psychosocial Processes, and Genetic Factors in BED

Emotional and social processes fluctuate in patients with BED during the course of the disease [96].

Many working strategies have been used to try to solve the problem from the point of view of emotions [26,103,104,105,106,107,108,109,110,111]. Nevertheless, the evidence on the causal relationship between negative feelings and binge eating is not conclusive, although it seems to be a maladaptive strategy for the regulation of emotions, such as substance abuse or self-harm [3,104,105]. In addition, the scientific community is increasingly interested in analyzing and intervening in the regulation of mental processes during binge eating episodes and also in the hours and days following them [103,112,113,114,115,116], so that, on many occasions, pharmacological and psychological interventions are combined to provide a solution to the problem [117,118]. Or, co-adjuvant therapies that provide added benefits, such as programming a formal physical activity, and are adapted to the needs of the patient [118] and/or cultivating the spiritual commitment as a way to reduce stress can improve the adjustment of emotions and regulation of the way to feed oneself [119,120,121].

More specifically, the therapies that have been most investigated to deal with the problem are cognitive–affective and cognitive–behavioral therapies. Both have been shown to significantly decrease the number of binge eating episodes in BED and obtain greater rates of abstinence during treatment, up to 6 months later, independently of the sociodemographic factors of the patients [122,123,124,125,126,127], but complete success rates have not been obtained, so further research on the multifactorial relationships is still needed.

Also, socio-cultural factors play a role in the development and manifestation of EDs, especially how communication media and social networks promote the idea that being thin is more attractive [128,129,130,131,132]. It has been demonstrated that body dissatisfaction is mediated by gender in binge eating behavior [133], and the factors specific to sexual minorities (for example, the stress of minorities and the connection with the lesbian, gay, bisexual, transgender, and queer community) should be taken into account, as they can affect binge eating in women of sexual minorities [134].

Very few studies were found that demonstrated that socioeconomic disadvantages can be an important risk factor in ED, especially with underlying genetic risks. The differences in sex are very important due to the different patterns of development (different activation of the genetic influence during adrenarche, puberty, etc.) [135,136].

Studies regarding the relationship between the environment and genetic factors on the influences between parents and children in feeding behavior have begun, even though there are no concluding data [137,138]. Twin studies indicated that these associations were largely due to shared genetic and/or environmental factors, rather than pure socialization effects unchanged by the stages of development [139].

A study that compared the relationship between the genotype 5-HITLPR and the style of parental relations did not show that this profile modulated the relationship between parental behavior and the appearance of ED [139], which continues to show the great variety of factors that influence these problems. For all these reasons, pioneering studies have begun to analyze genetic and environmental factors, such as the Eating Disorders Genetic Initiative (EDGI) [140], which is expected to provide us with more information in the future.

#### Strengths and Limitations

The first strength of the present review is the topic of analysis due to the need to continue research on BED, given its increased prevalence in Western society, as compared with AN and BN.

The completion of this study by a multi-professional team is a strength, as each reviewer provided information based on his or her professional and personal experiences. Having an external reviewer to provide a different perspective on research was useful for decreasing bias and increasing the reliability of this research.

As for the limitations, only English-language articles were reviewed. Also, articles from non-academic journals were not reviewed. We also did not consider doctoral theses that could have provided a view of emergent studies in the field. A limitation that could be a strength at the same time is the addition to the review of binge episodes produced by other EDs and not only BED, in order to broaden the information that related genetic and behavioral patterns present in BED. Lastly, most of the studies conducted provided data on Western countries, leaving out populations in Africa, Latin America, and Asia.

## 5. Conclusions

This systematic review is the first to explore the physiopathology of BED from two aspects: genetic and microbial, as well as its possible relationship with psychosocial factors. In general, genetic polymorphisms directly related to BED were observed in various genes (ABCA1, BDNF, DRD3, DRD4, FTO, and Taq1A), while other, non-direct relationships were also observed, although shared with other EDs, such as binge eating episodes (CLOCK, GLUT2, FABP2, HTR2A, MC4R, TAST1R2, and TAST1R3).

Also, a relationship was observed between dysbiosis in intestinal microbiota and BED. More specifically, a decrease in Akkermansia and an increase in Bifidobacterium, as compared with healthy subjects, were observed. Nevertheless, aside from the lack of studies on this aspect, recent studies support the idea of working on the ecosystem as a whole, avoiding segmentation.

More research needs to be conducted in this field. This is especially needed at the interdisciplinary level, as the results from the few studies conducted until now show that sociodemographic and psychosocial factors could be important risk factors for the development of EDs, especially if a baseline genetic risk exists.

To summarize, knowing the genetic factors, the microbiota function, and psychosocial factors of each patient could be important to preserve and improve their balance when confronting this disease. This systematic review highlights the important network of existing relationships between the genetic factors, intestine, and psychosocial factors in BED. Future investigations could provide more information regarding the mentioned relationships, and it could even offer a way to identify each patient, to be able to design effective therapy.

## Figures and Tables

**Figure 1 healthcare-12-01441-f001:**
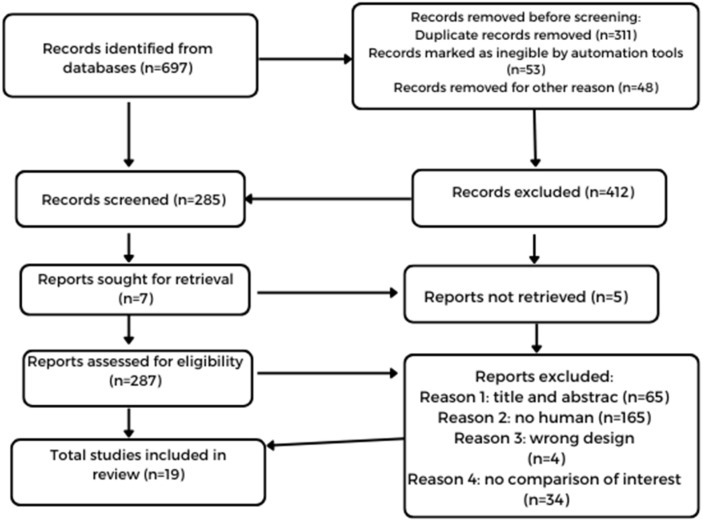
Flow diagram of identification and selection process via database and records of genes and BED.

**Figure 2 healthcare-12-01441-f002:**
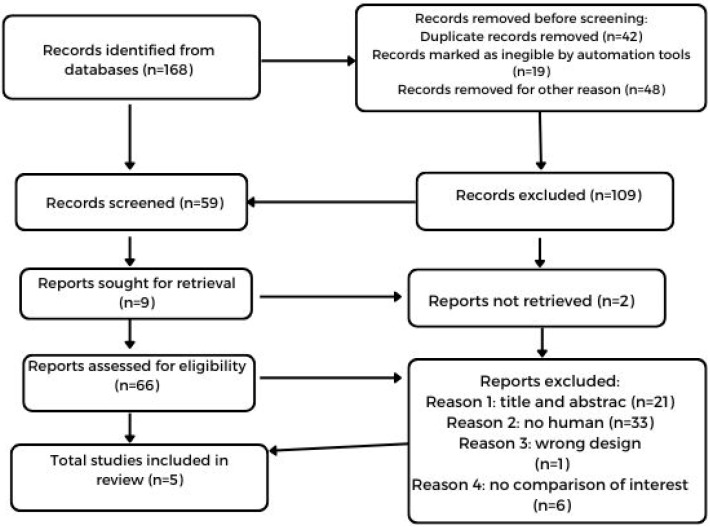
Flow diagram of identification and selection process in the databases and records of microbiota and BED.

**Figure 3 healthcare-12-01441-f003:**
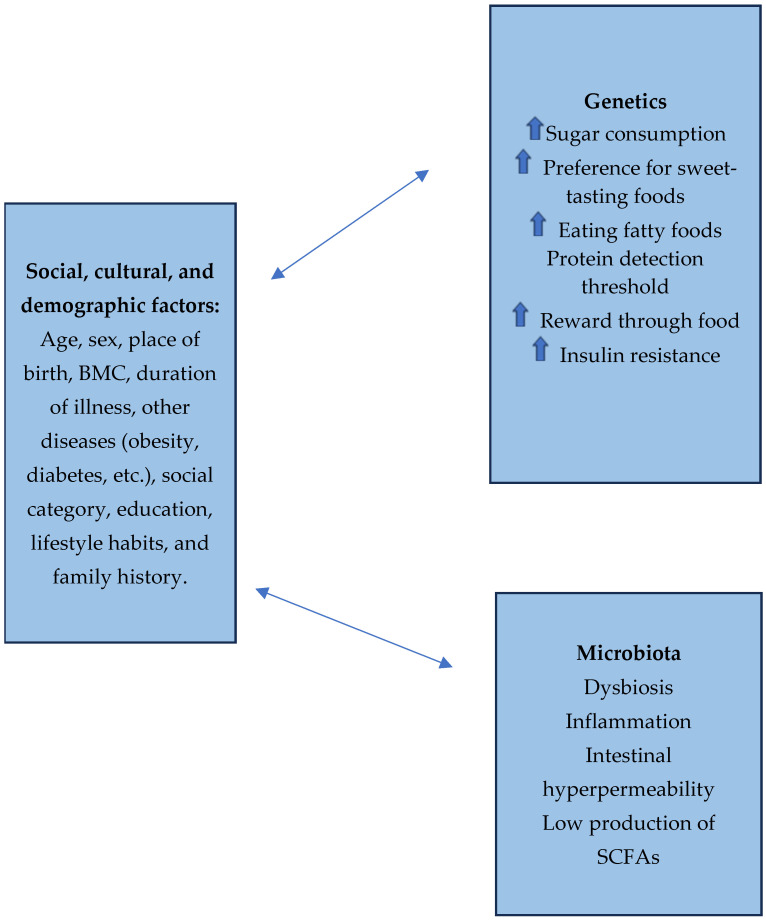
Flow diagram of selection process of genes, microbiota, sociodemographic factors, and BED. The arrow means high levels.

**Table 1 healthcare-12-01441-t001:** Specific information about polymorphisms and BED in the last 12 years.

Author(s), (Year)	Method (Design, Sample, and Measures) and Type of Analysis	Candidate Genes and Polymorphisms Studied	Main Conclusions	STROBE Score
Davis et al. (2012) [30]	Design: Cross-cutting. Sample: 230 obese adults (171 women and 59 men) (79 with BED). Measurements: DSM-IV-TR; BEQ; PFS; DEBQ-C; FCQ-T; genotyping. Analysis: ANOVA to evaluate the differences in genotype for the variable’s dependence on the sub-phenotype.	DRD2 rs1800497, rs1799732, rs2283265, rs12364283, rs6277	With the same weight, subjects with BED were significantly associated with rs1800497 and rs6277. BED presented less probability of carrying the minor T allele of rs2283265.	17
Munn-Chernoff et al. (2012) [31]	Design: Cross-cutting. Sample: 135 families Data: Family-based association tests; WSCB; BES; genotyping. Analysis: Haploview 27 to determine linkage disequilibrium (LD) patterns between the genetic variants. The Hardy–Weinberg equilibrium (HWE) was calculated using Merlin. Family-based association tests operated in FBAT 29 to perform all the association analyses.	5-HTTLPR SLC6A4 rs12945042 rs25531 rs6354 rs2020942 rs140700 rs2054847 rs1042173	No significant connections were found among any genetic variant and BED. It suggested that the use of polymorphisms in and near SLC6A4, including 5-HTTLPR, to identify genetic risk factors in TCA was useless.	20
Koren et al. (2015) [32]	Design: Cross-cutting. Sample: 1533 twin women. Measurements: DSM-IV; genotyping; BMI. Analysis: Logistic regression to examine the association between the concerns about weight/shape, binge eating and compensatory behaviors, and SNP.	HTR2A rs1923882 rs6561333 rs2296972	rs6561333 and rs2296972 showed protective influence against binge eating. The analyses provided preliminary evidence of intronic SNP in HTR2A and its association with binge eating.	21
Micali et al. (2015) [33]	Study: Cross-cutting. Sample: 4.916 subjects obtained from the longitudinal Avon study of fathers and sons aged 14 years (n = 5.958) and 16 years (n = 4.946). Data: DSM-V; BMI; genotyping. Analysis: Logistic regression model for association between SNP, BMI and binge eating.	FTO rs1558902	Association between binge eating and rs1558902. Significant relationship between BMI and binge eating.	21
Leehr et al. (2016) [34]	Study: Cross-cutting. Sample: 69 participants classified into three groups (obese with BED, obese without BED, healthy controls). Data: DSM-; BMI, -questionnaire on antisaccade impulsivity; genotyping. Analysis: One-way analyzes of variance (ANOVA) and Kruskal–Wallis tests were used to compute group differences in the sample characteristics. Pearson’s correlation for relationship between impulsivity measures.	COMT Val(108/158)	BED showed greater trait and behavioral impulsivity. BED + COMT Met/Met major deficits in inhibitory control.	18
Steiger et al. (2016) [35]	Study: Cross-cutting. Sample: 183 women with binge eating and purges (66.1% BN and rest TANE) Data: DSM-V; BMI, Barrat impulsiveness scale; Center for Epidemiological Studies–Depression CES-D; DAAP-BQ; genotyping. Analysis: Separate logistic multiple regressions, structured to explore the main and interaction effects of genetic factors on dependent variables.	DRD2 Taq1A DRD47R COMT	Gene combinations that codified for low levels of dopaminergic neurotransmission related to sensitivity to substance abuse in binge eating patients. DRD47R interacted with COMT to predict greater probability of substance abuse.	19
Micali et al. (2017) [36]	Design: Cross-cutting. Sample: 394 women with EDs; 4708 controls without EDs. Measurements: DSM-IV; BMI; PBI (GDP?); genotyping. Analysis: Logistic regression model for SNP haplotype association analysis and for the effect of genotypes OXT-R.	OXTR rs2254298 rs53576	OXTR variation was associated with excess eating, preference for sweet and fatty foods, and sensitivity to reward and punishment. Relationship between excess eating and low basal levels of OXT. Significant associations between the haplotype G-T-A-G and preference for sugar/fats. An allele from SNP rs53576 was negatively correlated with binge eating/purges behavior. The genotype GG had a greater risk of participating in binge eating and purges.	20
Palacios et al. (2018) [37]	Design: Cross-cutting. Sample: 50 obese adults (25 with EDs and 25 without EDs). Control group: 100 individuals with normal BMI. Measures: BED diagnostic (DSM-V), BMI, genotypes. Analysis: Deviations from Hardy–Weinberg equilibrium and the differences in the allele and genotype frequencies between groups were assessed by X2 tests. Multinomial logistic regression analysis was used to determine if a specific genotype is associated with obesity, with and without BED.	ANKK1 rs4938013 rs1800497	ANKK1 exhibited a possible relation with the dopaminergic system. The polymorphism rs1800497 (Taq1A) in the 8 axon demonstrated an association with BED.	19
Cameron et al. (2019) [38]	Design: Cross-cutting. Sample: 178 patients (73 BED with obesity, 55 BED without obesity, 50 with normal weight without BED). Measures: Weight, BMI, and fat percentage; EDE; ASQ: genotyped. Analysis: comparison of the genotypes of the SNP of FTO in the three groups. Chi-squared to compare anthropometric variables between groups. Multiple linear regression with binge frequency as dependent variable and the genotype, the level of ASQ, and the genotype x level of the ASQ scale as independent variables.	FTO rs1421085 rs1121980	The SNPs related to obesity and TCA also interacted with anxious attachment. Greater frequency of binge eating by those that had these polymorphisms.	20
Palmeira et al. (2019) [39]	Design: Cross-cutting. Sample: 93 women overweight or with obesity (31 with BED and 62 without BED). Measures: Anthropometry (weight, height, BMI); EDE; DSM-V; BES; genotyped. Analysis: Genotype frequency and allele through counting of genes and the balance of Hardy–Weinberg. Logistical regression for the association between genotypes and BED. Anthropometric characteristics, biochemical data, and neuropsychological test scores between BED and control through Student’s t-test for parametric samples and Mann–Whitney U test for non-parametric samples.	FTO rs9939609 SLC6A4 DRD2 rs1800497 BDNF rs6265 y rs16917237 GHRL rs696217 y rs4684677	There was no significant relationship between the SNPs analyzed and BED. In contrast to AN and the risk of obesity of allele A FTO, rs9939609 did not have potential relevance to BED.	20
Genis-Mendoza et al. (2020) [40]	Study: Cross-cutting. Sample: 99 teenagers with EDs. Data: DSM-V; QEWP-5; MINI-Kid; genotyped. Analysis: Differences between the leptin concentrations were analyzed with the help of Student’s *t*-test or ANOVA. The gene–gene interaction was analyzed with the help of general estimation equations.	FTO rs9939609 ABCA1 rs9282541	A relationship existed between rs9939609 and leptin. Individuals with BN and people with allele A in the variants FTO and ABCA1 showed consistently the highest log-leptin. The patients with AN or BED showed a tendency to this difference, but without statistical significance.	18
Yagin et al. (2020) [41]	Design: Cross-cutting. Sample: 180 women premenopausal with overweightness and obesity Measures: BES; BMI; biochemical measures (serum levels of leptin, insulin, and orexin A), genotyping. Analysis: Chi-square test to compare genotypes FAAH 385 C/A between women with and without BED. To analyze the non-parametric quantitative variables, the Mann–Whitney U test was used. Logistic regression model for predicting association of appetite regulators with risk of BED.	FAAH Genotipo FAAH 385 C/A rs324420	Women with BED had significantly higher levels of AEA, 2-AG, leptin, and insulin in comparison with women without BED. AEA, leptin, and insulin were the predictors of BED with a higher frequency of the allele A of the gene FAAH in women with BED.	19
González et al. (2021) [42]	Design: Cross-cutting. Sample: 324 patients (210 AN, 80 BN, and 34 BED). Measures: DSM-V; EDI-2, SCL-90R, and genotyping. Analysis: Student/Mann–Whitney or ANOVA/Kruskal–Wallis tests to differentiate between quantitative variables. Logistic regression model to analyze the association of a single marker. Age-adjusted logarithmic probability ratio tests for gene–gene integration.	DRD2 A2/A1; DRD3 Ser9Gly; DAT1 10R/9R.	No direct associations between these genes for BN and BED were found. The patients with AN that carried the genotype Ser9Gly showed higher risk of BED.	20
Heidinger et al. (2021) [43]	Design: Cross-cutting. Sample: 99 women patients with overweightness/obesity (72 with BED; 27 without BED). Measures: No. of days of binge eating in the last 28 days (frequency); weight, BMI and body fat percentage, and genotyping. Analysis: ANOVA for statistical differences. Chi-square test for comparison of genotypic frequencies. *T*-tests for independent samples for association between anthropometric variables and risk genotypes.	DRD2/ANKK1 Taq1A COMT MAO-A DAT-1	No significant relationship between BED and analyzed polymorphisms. Difficulty in observing group data in BED due to its complex aetiopathogenesis.	21
Magno et al. (2021) [44]	Design: Cross-cutting. Sample: 70 women with BMI between 40 and 60 kg/m^2^. Data: EVA; BES; 3-day dietary records; TaqMan assays^®^, and genotyping. Analysis: Multiple linear regression to compare influential models in MCR4. Graphical analysis of ordinary least squares fitted model residuals was performed to confirm their randomness.	MCR4 rs17782313	Higher prevalence of severe binge eating in >50% with at least one risk allele. MC4R rs17782313 related to ghrelin release, including feelings of hunger and satiety. More than half of the women with this polymorphism presented severe binge eating.	19
Ceccarini et al. (2022) [45]	Design: Transversal. Sample: 568 experimental group (332 AN, 122 BN, and 132 BED) and 172 control group. Measures: Anthropometry (weight, height, and BMI); DSM-V; genotyped. Analysis: Hardy–Weinberg for genotype frequencies. Chi-squared test for differences between genotypes and frequencies of alleles. ANOVA to find the influence of BMI in the genotype.	DRD2 rs6277 DRD4rs936416 ANKK1	Significant associations with BED from SNP DRD4rs936461. Specific combinations of variations in the genes DRD2 and DRD4 were factors of predisposition in TCA and to some psychopathological characteristics associated with AN, BN, and BED.	19
Nonino et al. (2022) [46]	Design: Cross-cutting. Sample: 177 patients undergoing bariatric surgery with subsequent weight gain and divided into two groups according to BED diagnosis. Measures: Anthropometry (weight, height, BMI, and BES) genotyping. Analysis: Student’s *t*-test was used to analyze the difference between groups. The Hardy–Weinberg equilibrium calculation was used to evaluate if allele and genotype frequencies in a population will remain constant from one generation to the next in the absence of disturbing factors. Logistic regression analysis was performed to verify the relationship between the studied polymorphisms and BED.	BDNF rs6265 DRD2 rs1800497	The presence of rs1800497 in the DRD2 gene and rs6265 in the BDNF gene, isolated and/or combined, showed additional risks for the development of BED in patients with obesity. Greater significance in the context of weight recuperation.	19

ASQ: Ages and Stages Questionnaire; BEQ: Berkeley Expressivity Questionnaire; BES: Body Esteem Scale; BMI: Body Mass Index; CES-D: Center for Epidemiologic Studies Depression Scale; DAAP-BQ: Dimensional Assessment of Personality Pathology–Basic Questionnaire; DEBQ-C: Dutch Eating Behavior Questionnaire for Children; DSM-V: Diagnostic and Statistical Manual of Mental Disorders; EDE: Eating Disorder Examination Questionnaire; EDI-2: Eating Disorders Inventory; EVA: analogical visual scale; FCQ-T: Food Craving Questionnaire–Trait; MINI-Kid: Mini-International Neuropsychiatric Interview for Children; PFS: Power of Food Scale; PBI: Parental Bonding Instrument; QEWP-5: Questionnaire on Eating and Weight Patterns–5; SCL-90R: Derogatis Symptom Checklist, Revised; WSCB: Warwickshire Health and Wellbeing Board. ABCA1: ATP-Binding Cassette Subfamily A member 1; ANKK1: Ankyrin Repeat and Kinase Domain Containing 1; BDNF: Brain-Derived Neurotrophic Factor; COMT: Catechol-O-Methyltransferase; DAT-1: Sodium-Dependent Dopamine Transporter; DRD2: Dopamine Receptor D2; DRD4: Dopamine Receptor D4; FAAH: Fatty Acid Amide Hydrolase; FTO: Fat Mass and Obesity-Associated Gene; GHRL: Ghrelin and Obestatin Prepropeptide; HTR2A: 5-Hydroxytryptamine Receptor 2A; MAO-A: Monoamine Oxidase A; SLC6A4: Solute Carrier Family 6 Member 4; MCR4: Melanocortin 4 Receptor; OXTR: Oxytocin Receptor.

**Table 2 healthcare-12-01441-t002:** Information about the method and analysis utilized in microbiota and BED research.

**Author(s), (Year)**	**Method (Design, Sample, Measurements) and Type of Analysis**
Bretón et al. (2016) [47]	Design: Experimental Sample: Women (29 healthy, 24 restrictive AN, 29 BN, and 13 BED) Measures: EDI-2, blood sample, and MADRS Analysis: Correlation analysis, plasmatic levels of IgG anti-ClpB, IgM, and IgG reactive to α-MSH in the same patients with DE and in the healthy group of comparison, their analysis was available from the previous study
Raevuori et al. (2016) [48]	Design: Transversal Sample: 1592 patients with EDs. Controls, 6368 general population Measures: CIE-10; DSM-V; and medication intake records Analysis: Conditional logistic regression model to analyze differences in age, sex, and place of residence with prevalence of prescription between patients and control individuals A linear model of mixed effects was used to investigate the DDD five years before the start of the treatment for eating disorders. As variables, the age (in years), the sex (male, female) and group of patients (controls, AN, BN, and BED) and the strata of control patients (a patient and four controls) as an aleatory effect were used Design: Transversal Sample: 150 women Measures: Magnetic resonance (MRI), anthropometry, YFAS, and stool sample Analysis: Partial-least-square regression for discrimination analysis was carried out (sPLS-DA) to analyze the microbiome data ANOVA calculated the importance of the differences in the metric of alpha diversity, the phylogenic diversity of Faith, Chaol, and the Shannon diversity index. The association of microbial genera with or without AF was evaluated using DESeq2 in R Design: Cohort Food4Gut Sample: 101 patients with obesity and binge eating Measures: Anthropometry; plasmatic biology; intestinal microbiota; arterial pressure; and EED-Q Analysis: Distributional analysis to evaluate the effect of the use of probiotics in the response variables (FA and BED) and in other explicative variables (age, weight, and IMC) for the same individual during the period analyzed Design: Experimental Sample: 44 obese patients after bariatric surgery (division of control group and experimental group) Measures: BED diagnostic (BES) and addiction to food (YFAS) Analysis: Distributional regression to evaluate the effect of the use of a probiotic in the response variables (FA and BED), and in other explicative variables (age, weight, and IMC) for the same individual during the period analyzed
Dong et al. (2020) [49]	Design: Transversal Sample: 150 women Measures: Magnetic resonance (MRI), anthropometry, YFAS, stool sample Analysis: Partial-least-square regression for discrimination analysis was carried out (sPLS-DA) to analyze the microbiome data ANOVA calculated the importance of the differences in the metric of alpha diversity, the phylogenic diversity of Faith, Chaol, and the Shannon diversity index. The association of microbial genera with or without AF was evaluated using DESeq2 in R
Leyrolle et al. (2020) [50]	Design: Cohort Food4Gut Sample: 101 patients with obesity and binge eating Measures: Anthropometry; plasmatic biology; intestinal microbiota; arterial pressure; and EED-Q Analysis: Distributional analysis to evaluate the effect of the use of probiotics in the response variables (FA and BED) and in other explicative variables (age, weight, and IMC) for the same individual during the period analyzed
Carlos et al. (2022) [51]	Design: Experimental Sample: 44 obese patients after bariatric surgery (division of control group and experimental group) Measures: BED diagnostic (BES) and addiction to food (YFAS) Analysis: distributional regression to evaluate the effect of the use of probiotic in the response variables (FA and BED), and in other explicative variables (age, weight, and IMC) for the same individual during the period analyzed

CIE-10: International Classification of Diseases; ClpB: Caseinolytic Peptidase B protein homolog; YFAS: Yale Food Addiction Scale.

**Table 3 healthcare-12-01441-t003:** Information about the type of intervention, results, and strobe score in the microbiota and BED research studies analyzed.

Author(s), (Year)	Type of Intervention	Results	STROBE Score
Bretón et al. (2016) [47]	Produced CLpB protein by *Escherichia coli* CLpB plasma protein concentration, α-MSH, of anti-ClpB IgG, IgM, and α-MSH-reactive IgG plasma levels	Elevated concentrations of CLpB in patients with TCA without differences in between subgroups. Positive and significant correlation between the concentrations of CLpB and EDI-2	Not applicable
Raevuori et al. (2016) [48]	To analyze the use of microbiota medicaments in a cohort of patients treated for BED, BN, and AN during the period of 5 years before the disorder	The individuals with BN and BED had received antimicrobial medicament prescriptions with greater frequency in comparison with their control (however, without significant difference) From the main medicament categories, the respective pattern was observed in the antibacterial and antifungal medicaments	20
Dong et al. (2020) [49]	Focus on system biology to demonstrate associations between physical activity and changes in the gut–brain–microbiota axis through fecal and metabolic microbes analysis, and cerebral data of anatomic connectivity.	The results of the study indicated a strong negative association between *Bacteroides*, *Akkermansia*, and *Eubacterium* with the AF Addiction to food was associated with intestinal dysbiosis characterized by an overabundance of *Megamonas* and reductions in *Akkermansia*, *Eubacterium*, and *Bacteroides*	20
Leyrolle et al. (2020) [50]	Non-targeted metabolomics	Decrease *Akkermansia* and *Itestinimonas* and increase *Bifidobacterium*, *Roseburia*, and *Anaerostipes* in obese patients with BED	21
Carlos et al. (2022) [51]	Five thousand million strains of Acidophilus and five thousand million strains of Bifidobacterium lactis (Bi-07) in a pill over 5 days	Before the surgery, a third of the patients presented an addiction to foods and binge eating diagnosis. The number of symptoms of YFAS and the punctuation BES significantly decreased in both groups in T1 in comparison with T0. However, a significant effect of the probiotic treatment was observed 1 year after the surgery (T2). Both the amount of symptoms of food addiction and the amounts of binge eating were lower in the probiotics group than in the placebo group (*p* = 0.037 and *p* = 0.030, respectively)	Not applicable

## Data Availability

Data are contained within the article.

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
