# Peer review of "Relationship of Genetic Polymorphisms and Microbial Composition with Binge Eating Disorder: A Systematic Review"

_healthcare, 2024, doi:10.3390/healthcare12141441_

Round 1

Reviewer 1 Report

Comments and Suggestions for Authors

General comments:

This is indeed an interesting topic addressed by the authors! The manuscript would be improved by professional editing in terms of academic writing (syntax, grammar, expression), as, in many cases, it is hard to understand.

Introduction: There is valid but mixed information here, thus not clearly supporting the need for your study. Also, you refer to all EDs now and then, when you actually focus on BED.  It is proposed to focus start from more general information on the subject (here you can refer to all Eds if you like), then starting to raise the issue (existing literature on the topic you wish to discuss – i.e. genes including relevant systematic reviews) and then highlight the gap in the literature that will lead to your study’s aim. You may focus less on detail regarding what Eds are and more on the fact that the existing therapy does not work. You currently provide one sentence stating that (lines 63 – 63) when this is useful for supporting the need for your study. You may also need to focus on just BED at some point and provide why it is important to provide therapy for people with BED, and what are the consequences for them.

Discussion: You need to present your findings in brief in the first paragraph in relation to your study aim. Then, you need to discuss each finding in a separate paragraph in comparison to other’s studies findings and try to comment in the realm of promoting research a step forward. Use previous data on other EDs only when there are no data for BED or in case you need to support an argument, as EDs in general were not part of your aim. Also, the combination of genes, BED and microbiota is not clearly discussed. Last, you need to include a paragraph with the strong points and areas for improvement of your study just before conclusions.

Conclusions: Here you should not discuss anything, just provide the basic conclusions of your study, what your manuscript has added to already existing knowledge, as well as a paragraph proposing what future research would be. A last but important thing is, does the evidence you collected strongly support the provision of dietary guidance? Avoid overstatement of your results.

Specific comments:

1.        Please take a look at the sentence lines 33 – 36 and rephrase, as it is hard to understand. Syntax seems also problematic in this sentence.

2.        The same applies to the following sentence lines 36 – 38. Which is a common intrinsic component?

3.        Who are they in line 45? (In addition, they may be under-diag-45 nosed.) You may refer to people with BED, but in terms of syntax this needs to be changed.

4.        Line 49 what is it? (it characterized…)

5.        There is not a verb in this sentence “Not being sufficient for the 63 decrease of incidence or prevalence.” Lines 63 – 64.

6.        What do you mean by “corporal weight”?

7.        Sentence found in 73 – 77 lines, is not clear. Please rephrase and/ or split into more sentences.

8.        This is not understood …”as Eds have factors derived from social interactions (familiar and similar groups of people) which affect both the present and how the person develops until it becomes an adult.”  What is the present that precedes adulthood? Lines 87 – 89.

9.        Do you mean improve instead of better in the following? : and so try and better the strategy in the field of nutrition for their treatment.” Lines 108 – 109

10.   2.2. avoid repeating the key word binge eating disorder. Rephrase to have this word as the first keyword plus one of the following. (this is an idea, you may re-arrange it differently). Also, if your aim is to include studies on BED or compulsive eating, this should be also depicted on your keywords.

11.   Line 128 filters were applied not made

12.   The criteria were followed line 130

13.   2.3 The eligibility criteria had all of them to be met or just one? Please clarify that.

14.   Line 231 Tabla,

15.   Table 2 Type of intervention

16.   Table 2. The title should depict the information included, i.e. Methodological characteristics and outcomes of studies included. Also, what “it doesn’t proceed means”. Strobe score might be just presented in your text and not on table to give more space on results and methods. Finally, consider having 2 tables, one with methods and one with results.

17.   Figure three should be self-explanatory. It is not clear what the parameters presented under the binge eating disorder subgroups are!

18.   You need to state in a better way your aim in the discussion section. Information on what?

19.   4.1. The first paragraph in your discussion section needs to present your findings!

20.   Line 307 is this major instead of mayor?

21.   Lines 412 – 116 are not understood. Please rephrase.

22.   Is fiver fiber in line 522?

Comments on the Quality of English Language

The manuscript would be improved by professional editing in terms of academic writing (syntax, grammar, expression), as, in many cases, it is hard to understand. There are also typos in words. I provided some in the detailed comments, but you need to thoroughly check you rmanuscript.

Author Response

This is indeed an interesting topic addressed by the authors! The manuscript would be improved by professional editing in terms of academic writing (syntax, grammar, expression), as, in many cases, it is hard to understand.

  • Thank you very much for your comments. We have taken the suggestions into account and the manuscript has been revised by a professional translator.

Introduction: There is valid but mixed information here, thus not clearly supporting the need for your study. Also, you refer to all EDs now and then, when you actually focus on BED.  It is proposed to focus start from more general information on the subject (here you can refer to all Eds if you like), then starting to raise the issue (existing literature on the topic you wish to discuss – i.e. genes including relevant systematic reviews) and then highlight the gap in the literature that will lead to your study’s aim. You may focus less on detail regarding what Eds are and more on the fact that the existing therapy does not work. You currently provide one sentence stating that (lines 63 – 63) when this is useful for supporting the need for your study. You may also need to focus on just BED at some point and provide why it is important to provide therapy for people with BED, and what are the consequences for them.

  • Thank you for your recommendations. We have modified and ordered some paragraphs for its better understanding:
    • Paragraph 1: general description of ED.
    • Paragraph 2: justification for the fluctuation between different EDs and their need in some studies to be analyzed together;
    • Paragraph 3: prevalence of ED among the population highlighting BED.
    • Paragraph 4: current treatment of BED.
    • Paragraph 5: use of GWAS to look for the causes of the disease in genetic factors.
    • Paragraph 6: addition of the microbiome study to complement the information from GWAS, especially the BEGIN study.
    • Paragraph 7: Justification of the need for further research in multidisciplinary studies to improve the understanding of the TCA, especially BED.
    • Paragraph 8: Specific objectives of our review.

Discussion: You need to present your findings in brief in the first paragraph in relation to your study aim. Then, you need to discuss each finding in a separate paragraph in comparison to other’s studies findings and try to comment in the realm of promoting research a step forward. Use previous data on other EDs only when there are no data for BED or in case you need to support an argument, as EDs in general were not part of your aim. Also, the combination of genes, BED and microbiota is not clearly discussed. Last, you need to include a paragraph with the strong points and areas for improvement of your study just before conclusions.

  • Thank you for your recommendations. The main findings have been added at the beginning of the discussion in relation to the objective of the study.
  • We have also improved the wording with respect to the objective of the study.
  • Paragraphs that we felt might confuse the reader have been removed. For example, line 314, line 496 (item 4.3 has been deleted and replaced by 4.4).
  • There are no studies yet that combine Genes, microbiota and BED. The BEGIN study described in the introduction is pioneering and recent (no published results).
  • The section on strengths and limitations has been introduced at the end of the discussion. Line 538

Conclusions: Here you should not discuss anything, just provide the basic conclusions of your study, what your manuscript has added to already existing knowledge, as well as a paragraph proposing what future research would be. A last but important thing is, does the evidence you collected strongly support the provision of dietary guidance? Avoid overstatement of your results.

  • Thank you for your correction. We have modified the conclusions only to the results obtained from our review.

Specific comments:

  1. Please take a look at the sentence lines 33 – 36 and rephrase, as it is hard to understand. Syntax seems also problematic in this sentence.
  • The sentence has been modified. Line 43
  • The main characteristics of each disorder have also been introduced at the request of reviewer 2 in the previous paragraph.
  1. The same applies to the following sentence lines 36 – 38. Which is a common intrinsic component?
  • Currently, genetic and neurological factors are the most studied.

  1. Who are they in line 45? (In addition, they may be under-diag-45 nosed.) You may refer to people with BED, but in terms of syntax this needs to be changed.
  • From 2 to 4% of the western population.

  1. Line 49 what is it? (it characterized…)
  • This sentence has been moved to the beginning of the introduction. The word Characterizer has been replaced by "in the BED are presented peridoso recurrent....."

  1. There is not a verb in this sentence “Not being sufficient for the 63 decrease of incidence or prevalence.” Lines 63 – 64.
  • Reworded: However, they are not sufficient to reduce incidence and/or prevalence.

  1. What do you mean by “corporal weight”?
  • The word has been changed: Body weight. Line 80

  1. Sentence found in 73 – 77 lines, is not clear. Please rephrase and/ or split into more sentences.
  • Thanks for the clarification. It has been divided into two sentences

  1. This is not understood …”as Eds have factors derived from social interactions (familiar and similar groups of people) which affect both the present and how the person develops until it becomes an adult.” What is the present that precedes adulthood? Lines 87 – 89.
  • The sentence has been modified. In addition, it has also been reordered following the initial recommendations. Line 94
  1. Do you mean improve instead of better in the following? : and so try and better the strategy in the field of nutrition for their treatment.” Lines 108 – 109

- This sentence has been eliminated. As recommended by reviewers 1 and 2, the allusions to nutrition have been eliminated because they are not relevant to this article.

  1. 2. avoid repeating the key word binge eating disorder. Rephrase to have this word as the first keyword plus one of the following. (this is an idea, you may re-arrange it differently). Also, if your aim is to include studies on BED or compulsive eating, this should be also depicted on your keywords.
  • Following your instructions, changes have been added to lines121 to 126
    • (Binge Eating disorder OR compulsive eating) AND (genes OR genotypes OR polymorphism)
    • (Binge Eating disorder OR compulsive eating) AND (microbiome OR microbiota)
    • (Binge Eating disorder OR compulsive eating) AND (genes OR genotypes OR polymorphism) AND (psychological factors OR psychosocial development OR psychosocial factors)

  1. Line 128 filters were applied not made
  • Thank you for the clarification. It has been modified.

  1. The criteria were followed line 130
  • Thank you for the clarification. It has been modified.

  1. 3 The eligibility criteria had all of them to be met or just one? Please clarify that.
  • All of them had to be complied with. Specified in the text. Line 146

  1. Line 231 Tabla,
  • Thank you for the clarification. It has been modified.

  1. Table 2 Type of intervention
  • Thank you for the clarification. It has been modified.

  1. Table 2. The title should depict the information included, i.e. Methodological characteristics and outcomes of studies included. Also, what “it doesn’t proceed means”. Strobe score might be just presented in your text and not on table to give more space on results and methods. Finally, consider having 2 tables, one with methods and one with results.
  • The title of Table 2 has been modified.
  • Table 2 has been divided into: Table 2 (method and analysis) and Table 3 (type of intervention and results).
  • We considered leaving Strobe Score in Table 3 because there is already space in the table after it has been divided.

  1. Figure three should be self-explanatory. It is not clear what the parameters presented under the binge eating disorder subgroups are!
  • Thank you for your appreciation. The title has been modified: "Social, cultural and demographic factors".

  1. You need to state in a better way your aim in the discussion section. Information on what?
  • Thanks for the clarification. The aim of the present work consisted in giving information as clear and reliable as possible on the genetic polymorphisms and microbiota status related to BED through scientific bibliography for it to be of utility in the search of the professional inter-ventions in BED.

  1. 4.1. The first paragraph in your discussion section needs to present your findings!
  • Thank you very much for the suggestion. The main results have been included. Line 302-307.

  1. Line 307 is this major instead of mayor?
  • Thank you for your appreciation. Changed: major.

  1. Lines 412 – 116 are not understood. Please rephrase.
  • Thank you for your appreciation. The sentence has been modified for a better understanding.

  1. Is fiver fiber in line 522?
  • Yes, it is fiber. However, this paragraph has been deleted on the recommendation of reviewer 2.

Reviewer 2 Report

Comments and Suggestions for Authors

The review aimed to investigate genetic studies regarding the microbiota carried in humans and find relation with genetic make-up of patients diagnosed with eating disorders.

 Seems to be of interest.

Title: not clear if this is what meant: I rephrased it like this: Understanding the pathophysiology of binge eating disorders for a collaborative nutrition precision  treatment.

Abstract:

line 14: i guess "this" investigation & not "the"? The aim or objective needs to be focused on the title, the flow of ideas are not coherent.,

Introduction:

Line 87: Eds be consistent with ED ( eating disorder). Line 92 (they  & not (thy).

Line 89-95. this is a 1 sentence. plz re write, section it and clarify .

same to lines 96-100, 1 sentence.

Generally, you have frequent long sentence's that might mislead the reader)  ( Discussion: line 482-487 

 Introduction lacks information regarding the possible nutrition precision treatments as mentioned in title. plz Get more literature.

Materials and Methods; study protocol, I am not familiar with the PRISMA regulations. But I cannot find in the key word search ( nutrition ) or (nutrition precision).

Results section: very lengthy and many  abbreviations for genetic make ups.

I cannot find studies related to the review title..

page 13, Table 3: there is a confounding effect for post bariatric surgeries which is the hormonal effect of ghrelin  & not only the micobiota  effect.

Discussion: you suddenly shifted to the hormonal role in satiety and in hunger. However, you have not shown if any relation exerted by the micrbiota & these hormones. in some places you used ED & in other ( Ed s) unify the abbreviation.

Figure 3: it would be excellent if you explained earlier studies relating the food items & BN or any ED cannot find the resources.

Line 319-221: Taste Receptor 319 Type 1 Member 2 (TAST1R2) and Taste Receptor Type 1 Member 3 (TAST1R3 .

line 386: it cannot be said (when diseased with BN), it should be when diagnosed with BN.

this section is very length & will let the reader lose track & focus.

line 482-487: the 5 lines involve one sentence! re phrase & write.

discussion is very lengthy.

Line 522: correct "fiver". check what is the proper word.

Conclusion: is very lengthy, you have again detailed about the foods that will improve symbiosis. While in the initial paragraphs you have not detailed about food nor your literature referred to experimental or intervention studies  revealing precision nutrition and EB.

Your tile is not reflective of the review

line 527: 1.2 & not 1,2

Comments on the Quality of English Language

The manuscript writing needs comprehensive revision and editing in terms of the language, clarity, coherence and length of sentences. 

Author Response

The companion file has all the changes with tracked changes. The manuscript version 2 is the definitive one with the revision in English

The review aimed to investigate genetic studies regarding the microbiota carried in humans and find relation with genetic make-up of patients diagnosed with eating disorders.

 Seems to be of interest.

Title: not clear if this is what meant: I rephrased it like this: Understanding the pathophysiology of binge eating disorders for a collaborative nutrition precision  treatment.

  • We have modified the title so that it is not misleading. Also according to their recommendations at the end of this document as well as Reviewer 2: Relationship of genetic polymorphisms and microbial composition with Binge Eating Disorder. A systematic review

Abstract:

line 14: i guess "this" investigation & not "the"?

  • Thanks for the clarification.

The aim or objective needs to be focused on the title, the flow of ideas are not coherent.,

  • The title has been modified so that it is not misleading and the relationship with the objectives and results is clearer.

Introduction:

Line 87: Eds be consistent with ED ( eating disorder).

  • All incorrect Eds have been changed. ED has been homogenized

Line 92 (they  & not (thy).

- Changed. Thanks for the appreciation

Line 89-95. this is a 1 sentence. plz re write, section it and clarify.

  • The sentence has been modified (line 95): Deriving in acceptance of sensory characteristics of food/dishes; economic and ecological factors; perception they have of food and how they classify it; symbolic factors linked to it in relation to elements of social status, gender, age, beliefs, knowledge and assigned values. As well as the relationship with health, image and/or aesthetics [21,22].

same to lines 96-100, 1 sentence.

  • The sentence has been modified (line 100): Through the present review we propose a biological disease model based on contemporary genetic and microbiome status findings to which psychosocial aspects are added.

Generally, you have frequent long sentence's that might mislead the reader)  ( Discussion: line 482-487)

  • Thank you for your appreciation. The sentence has been modified (line 519): The role of sociocultural factors in the development and manifestation of ED has been extensively studied, in particular, how the media and social networks promote the idea that thinness is more attractive [130, 143,144,145,146].

Introduction lacks information regarding the possible nutrition precision treatments as mentioned in title. plz Get more literature.

  • The concept of precision nutrition has been eliminated as it is not relevant in this review. Likewise, the conclusions have been modified

Materials and Methods; study protocol, I am not familiar with the PRISMA regulations. But I cannot find in the key word search ( nutrition ) or (nutrition precision).

  • We have eliminated the concept of precision nutrition due to its lack of relevance in the text.

Results section: very lengthy and many abbreviations for genetic make ups.

  • Abbreviations are explained at the end of each table. We cannot omit them as they are important. We have reduced what is dispensable so that the information is correctly understood. We have followed your recommendations and reduced the discussion section as well as the conclusions.

I cannot find studies related to the review title..

  • Thank you for your appreciation. Also on the advice of reviewer 2 we have modified the title: Relationship of genetic polymorphisms and microbial composition with Binge Eating Disorder. A systematic review.

Discussion: you suddenly shifted to the hormonal role in satiety and in hunger. However, you have not shown if any relation exerted by the micrbiota & these hormones.

  • En el caso de la insulina, las referencias 60, 91 y 98 hablan de como los productos del metabilismo de la microbiota (butirato y propionato) se relacionan con la secrección de insulina, and …they are both related in investigations of binge eating and BED [97, 99].
  • Y, en la frase siguiente se establece una relación entre hormona, microbiota y proteinas de señalización genética ….adults with overweight [101] potentially due to the positive regulation of the signaling of the receptor GLP-1…

In some places you used ED & in other ( Ed s) unify the abbreviation.

  • ED has already been homogenized throughout the text.

Figure 3: it would be excellent if you explained earlier studies relating the food items & BN or any ED cannot find the resources.

- Since we are not going to add the nutritional factor, we consider that this point is no longer relevant.

line 386: it cannot be said (when diseased with BN), it should be when diagnosed with BN.

  • Thank you for your appreciation. It has been modified

this section is very length & will let the reader lose track & focus.

  • Paragraphs that were considered irrelevant have been summarized.
  • Point 4.3 has been eliminated because we considered it irrelevant for this revision. Thus, the size of this section has been reduced.

line 482-487: the 5 lines involve one sentence! re phrase & write.

  • The wording has been modified (line 519): Also, the role of sociocultural factors in the development and manifestation of ED has been extensively studied, in particular, how the media and social networks promote the idea that thinness is more attractive [130, 143,144,144,145,146].

discussion is very lengthy.

- The information has been reduced and clarified. The information regarding nutritional aspects has been eliminated considering your suggestions and those of reviewer 3 (not applicable in this revision).

Line 522: correct "fiver". check what is the proper word.

  • It has been modified. The correct word is dietary fiber. However, this information has been deleted from the conclusions section.

Conclusion: is very lengthy, you have again detailed about the foods that will improve symbiosis. While in the initial paragraphs you have not detailed about food nor your literature referred to experimental or intervention studies  revealing precision nutrition and EB.

  • Thank you for your appreciation. The conclusions have been modified according to the ideas of the text. The concepts of nutrition and precision nutrition have been eliminated as well as the wording in this regard.

Your title is not reflective of the review

  • Modified according to the information in the text: Relationship of genetic polymorphisms and microbial composition with Binge Eating Disorder. A systematic review

line 527: 1.2 & not 1,2

  • Thank you for your appreciation. It has been modified

Comments on the Quality of English Language

The manuscript writing needs comprehensive revision and editing in terms of the language, clarity, coherence and length of sentences. 

  • An exhaustive English revision has been carried out to improve English comprehension.

Reviewer 3 Report

Comments and Suggestions for Authors

Introduction:

Much of this introduction provides good information to the reader. However, it often gets lost in the odd vernacular and syntax that is used. I would highly recommend collaborating with a native English speaker to edit the manuscript as a whole.

A brief introduction around what characterizes the three disorders discussed when they are introduced would benefit the readability of the paper.

Line 32: Please distinguish which version of the DSM to which you are referring. This will help readers in the future after new versions have come out.

Line 33: Please consider rephrasing. Currently this statement regarding the differences in how the conditions present is hard to understand as it is currently written.

Line 36: Please give an example as to what you mean by “instigated by substances”

Line 74: I believe you are referring to the microbiome here. It would serve better to refer to it as the microbiome instead of microbes.

Line 74: Please use body weight instead of corporeal weight.

Why focus solely on BED when also discussing AN and BN throughout the rest of the introduction?

Discussion:

Line 297 genome association is already relayed in the acronym GWAS. So you do not need to include that again after it.

Did the studies reviewed locate any specific SNPs?

Line 301: Is BED a subtype of obesity or eating disorder? Those are two very different things. Obesity is a degree of being overweight or overfat versus BED refers to eating habits. You can have people that are both obese and also display BED patterns. But just because someone has BED does not mean they will be obese.

Line 326: GLUT-2 allows glucose to leave the enterocyte of the intestines into the bloodstream and to subsequently enter liver and kidney cells. It is not in charge of carrying glucose through the blood.

Conclusions:

Based on the lack of research on BEN and various aspects like genetics and microbiome diversity you point out throughout the discussion I find your recommendations on things such as those needing to be taken into account during treatment a bit inappropriate. As you point out, currently we lack much of the research needed for us to make recommendations based on those aspects and so a better call would be for researchers focus on those aspects than for practitioners to take them into account before we have the research to back it up.

Comments on the Quality of English Language

As stated with my comments. There is some good information presented throughout the paper. However, the way it is written makes it very difficult to comprehend at times. I would highly recommend working with a native english speaker to provide substantial english editing. 

Author Response

the companion file has all the changes with tracked changes. The manuscript version 2 is the definitive one with the revision in English

Introduction:

Much of this introduction provides good information to the reader. However, it often gets lost in the odd vernacular and syntax that is used. I would highly recommend collaborating with a native English speaker to edit the manuscript as a whole.

  • An exhaustive English revision has been carried out to improve English comprehension.

A brief introduction around what characterizes the three disorders discussed when they are introduced would benefit the readability of the paper.

  • Introduced line 33: AN is characterized by a restriction of energy intake relative to necessary re- quirements, significantly low body weight for the patient's age, sex, developmental stage, and physical health, fear of gaining weight, and distortion of reality regarding weight; BN consists of recurring episodes of binge eating (minimum 1 time per week for 3 months) followed by compensatory behaviors (vomiting, laxative and/or diuretic abuse, intense exercise, or fasting); BED is characterized by having recurring binge eating periods (≥1 times per week, for 3 months minimum), brief (≤2 hours), and distressing, during which the patients feel a loss of control over which they consume large quantities of food compa-red to the majority of people in similar circumstances [1].

Line 32: Please distinguish which version of the DSM to which you are referring. This will help readers in the future after new versions have come out.

  • Added: DSM-V

Line 33: Please consider rephrasing. Currently this statement regarding the differences in how the conditions present is hard to understand as it is currently written.

  • Changed line 43: In general they share an altered eating behavior [2],

Line 36: Please give an example as to what you mean by “instigated by substances”

  • Gracias por la sugerencia. Se ha añadido, línea 46. Además, los trastornos instigados por sustancias (consumo de alcohol, tabaco) son más prevalentes en pacientes con trastornos alimentarios que en la población general y viceversa, por lo que se asume que se trata de un componente intrínseco común ligado a la adicción [3].

Line 74: I believe you are referring to the microbiome here. It would serve better to refer to it as the microbiome instead of microbes.

  • Thanks for the appreciation. Changed: Microbiome.

Line 74: Please use body weight instead of corporeal weight.

  • Thanks for the suggestion. It has been modified. Line 80

Why focus solely on BED when also discussing AN and BN throughout the rest of the introduction?

  • There are few studies that analyze BED separately from the other ACTs. For this reason, and as relationships between them will be found in the articles analyzed, we consider it interesting to add a small review in the introduction.

Discussion:

Line 297 genome association is already relayed in the acronym GWAS. So you do not need to include that again after it.

  • Thank you for your appreciation, it has been removed. Line 317

Did the studies reviewed locate any specific SNPs?

- This sentence has been deleted at the request of reviewer 1.

Line 301: Is BED a subtype of obesity or eating disorder? Those are two very different things. Obesity is a degree of being overweight or overfat versus BED refers to eating habits. You can have people that are both obese and also display BED patterns. But just because someone has BED does not mean they will be obese.

- Changed. In general, BED manifests with obesity due to a propensity to binge eat that may be influenced by hyperreactivity to the hedonic properties of food.... Line 314

Line 326: GLUT-2 allows glucose to leave the enterocyte of the intestines into the bloodstream and to subsequently enter liver and kidney cells. It is not in charge of carrying glucose through the blood.

  • Changed: The Glucose Transporter 2 (GLUT2) protein allows glucose to leave the enterocytes of the intestine into the bloodstream and subsequently enter the cells of the liver and kidneys. Line 341

Conclusions:

Based on the lack of research on BEN and various aspects like genetics and microbiome diversity you point out throughout the discussion I find your recommendations on things such as those needing to be taken into account during treatment a bit inappropriate. As you point out, currently we lack much of the research needed for us to make recommendations based on those aspects and so a better call would be for researchers focus on those aspects than for practitioners to take them into account before we have the research to back it up.

  • Allusions to nutritional treatments have been eliminated both in the conclusions and in the title because they were inappropriate. The conclusions have been modified according to its recommendations.

Comments on the Quality of English Language

As stated with my comments. There is some good information presented throughout the paper. However, the way it is written makes it very difficult to comprehend at times. I would highly recommend working with a native english speaker to provide substantial english editing. 

  • Thank you for your appreciation. The document has been reviewed again by a native speaker.

Round 2

Reviewer 3 Report

Comments and Suggestions for Authors

Introduction:

Line 55: How does this 2-4% prevalence for BED compare to the prevalence of the other disorders discussed? Please add the prevalence of the other disorders so the readers can properly compare.

Line 65: I believe you are referring to the cognitive impairments related to being malnourished. Not being nourished.

Line 71: research or literature would be the correct terms here. Not bibliography

Line 84: The information regarding the BEGIN study do not seem relevant to the rest of the paper. Unless you are summarizing information that came about from that investigation, it is not relevant. Please remove it.

Line 92 and 93: Did you mean BED instead of ED?

Methods

Line 167: What diagnostic criteria changed regarding diagnosing via the DSM-IV vs the DSM-V

Results:

This section reads too much like an annotated bibliography. These sections need to be put into a more cohesive paragraph form telling the general story about what the main results were. Not just X study said this and Y study said this as it is currently written. If readers wanted to just know what each individual study found, they can just look at the provided tables.

Discussion:

Line 301: “an” not “and”

Line 302: The more appropriate term would be research or literature, not bibliography.

Line 312: Please add citations for the statements like “decreased symptoms of BED and the addiction to food”.

Line 331: Than not that

Line 347: It should state that those individuals have been found to be more likely to have an elevated sugar and UP food intake. With association studies such as these we can never say that just because someone has a certain gene or SNP that they will absolutely have a certain kind of behavior.

Line 364: The more appropriate term would be previous, not posterior.

Line 369: Did you mean BED instead of ED?

Line 387: Please add citations

Line 391: I believe you mean “gene” instead of “gen”

Line 426: The more appropriate term would be observational, not observatory.

Line 502: I am not sure what you mean by posterior hours and days. Consider rephrasing

Line 511: Where these results found in patients with BED?

Comments on the Quality of English Language

The english editing throughout this paper is much improved. There are a few minor points in which improper synonyms are used. But the paper reads significantly better now. 

Author Response

Introduction:

Line 55: How does this 2-4% prevalence for BED compare to the prevalence of the other
disorders discussed? Please add the prevalence of the other disorders so the readers can
properly compare.

- The requested information has been added. The wording has been reworded as
follows: Despite being less well researched, BED is the most prevalent in Western
society [5], with a prevalence ranging from 2% to 4% of the population aged 16-20
years, compared to AN (0.8-2%) and BN (2-3%)[4].

Line 65: I believe you are referring to the cognitive impairments related to being malnourished.
Not being nourished.

- Thank you for your appreciation. It has been modified.

Line 71: research or literature would be the correct terms here. Not bibliography
- Corrected

Line 84: The information regarding the BEGIN study do not seem relevant to the rest of the
paper. Unless you are summarizing information that came about from that investigation, it is not
relevant. Please remove it.
- We consider it important to mention this study because it is a recent and important
initiative in relation to the study of genetics (also microbial) with binge eating disorder
and, therefore, in later years will offer great advances in this regard. For this reason, we
have followed your indications and reduced the information regarding this study but
without eliminating it completely. The wording of the paragraph has been reworded as
follows: Thus, based on the two major studies mentioned above, the Binge Eating
Genetics Initiative (BEGIN) [19] emerged as a multidimensional study examining the
interac-tion of the genome with the gut microbiota, as well as phenotypic data to provide
treatment responses for BED and BN.

Line 92 and 93: Did you mean BED instead of ED?
- It is intended to refer to all ED. However, for better understanding we have made a
clarification in the wording: Despite advances in the field of biomedicine, the cause of
ED in general and BED in particular has not been fully explained. This is because ED
(and BED) have a high social interaction.......

Methods
Line 167: What diagnostic criteria changed regarding diagnosing via the DSM-IV vs the DSM-V
- The DSM-V has maintained the same characterization of binge eating disorder that
appeared in the DSM-IV research criteria. There is only one significant change from the
DSM-IV to DSM-V research criteria: the temporal criterion (criterion D). In DSM-IV-TR
the frequency of binge eating is at least 2 times per week in the last 6 months, and in
DSM-V both the frequency and the temporality required to meet the criterion is me - nor
(once a week for three months). The DSM-V also includes for binge eating disorder
specifications for remission and severity.

For the sake of clarity we have left only the DSM-V

Results:
This section reads too much like an annotated bibliography. These sections need to be put into
a more cohesive paragraph form telling the general story about what the main results were. Not
just X study said this and Y study said this as it is currently written. If readers wanted to just
know what each individual study found, they can just look at the provided tables.
- The wording has been modified. See lines 181-210 and lines 239- 275.

Discussion:
Line 301: “an” not “and”
- Corrected

Line 302: The more appropriate term would be research or literature, not bibliography.
- Corrected

Line 312: Please add citations for the statements like “decreased symptoms of BED and the
addiction to food”.

Introduced reference 52

Line 331: Than not that
- Corrected

Line 347: It should state that those individuals have been found to be more likely to have an
elevated sugar and UP food intake. With association studies such as these we can never say
that just because someone has a certain gene or SNP that they will absolutely have a certain
kind of behavior.
- The wording has been modified: Individuals with TT genotype for the rs5400 SNP of the
SLC2A2 gene, which codes for GLUT2, are more likely to have a higher sugar intake
and an increased likelihood of predisposition to ultra-processed foods, derived from
under-sensing of blood glucose [66,70,71]. However, there are no studies analyzing the
relationship between this gene and BED.

Line 364: The more appropriate term would be previous, not posterior.
- Corrected

Line 369: Did you mean BED instead of ED?
- No. ED is ok

Line 387: Please add citations
- Reference 41 has been added.

Line 391: I believe you mean “gene” instead of “gen”
- Yes. Corrected

Line 426: The more appropriate term would be observational, not observatory.
- Corrected

Line 502: I am not sure what you mean by posterior hours and days. Consider rephrasing
- Reworded: In addition, the scientific community is increasingly interested in analyzing
and inter-vening on the regulation of mental processes during the binge eating episode
and also in the hours and days following the binge.

Line 511: Where these results found in patients with BED?
- Yes. Has been added to clarify the sentence. Reworded as follows: Both have been
shown to significantly decrease the number of binge eating episodes in DBT and to
obtain higher abstinence rates during treatment.... 
